# CD46–ADC Reduces the Engraftment of Multiple Myeloma Patient-Derived Xenografts

**DOI:** 10.3390/cancers15225335

**Published:** 2023-11-09

**Authors:** Michael J. VanWyngarden, Zachary J. Walker, Yang Su, Olivia Perez de Acha, Brett M. Stevens, Peter A. Forsberg, Tomer M. Mark, William Matsui, Bin Liu, Daniel W. Sherbenou

**Affiliations:** 1Division of Hematology, University of Colorado Anschutz Medical Campus, Aurora, CO 80045, USA; vanwyngarden.mike@gmail.com (M.J.V.); zachary.j.walker@cuanschutz.edu (Z.J.W.); oliviaperezachalopez@gmail.com (O.P.d.A.); brett.stevens@cuanschutz.edu (B.M.S.); peter.forsberg@cuanschutz.edu (P.A.F.); tomer.mark@karyopharm.com (T.M.M.); 2Department of Anesthesia, UCSF Helen Diller Family Comprehensive Cancer Center, University of California, San Francisco, CA 94143, USA; 3Livestrong Cancer Institutes, Dell Medical School, University of Texas at Austin, Austin, TX 78705, USA; william.matsui@austin.utexas.edu

**Keywords:** multiple myeloma, patient-derived xenografts, CD46, antibody–drug conjugate, multiple myeloma-initiating cells

## Abstract

**Simple Summary:**

Multiple myeloma (MM) is incurable, implying that the disease cells are inherently resistant to current agents and inevitably lead to relapse. One strategy to improve the treatment paradigm is to pursue novel drug targets that are associated with relapsed disease. Towards this end, we developed a novel antibody–drug conjugate (ADC) that targets the cell surface complement inhibitor CD46. Here, we study the potential of this new agent to affect disease initiation. The gene encoding for CD46 is amplified on chromosome 1q in the majority of relapsed myeloma patients, as well as cells containing stem-like aldehyde dehydrogenase activity. We demonstrate the curative potential of CD46–ADC via its ability to abrogate disease engraftment of primary MM cell xenografts. In combination with a recent clinical trial, this study supports the continued study of CD46 as a therapeutic target in MM.

**Abstract:**

An antibody–drug conjugate (ADC) targeting CD46 conjugated to monomethyl auristatin has a potent anti-myeloma effect in cell lines in vitro and in vivo, and patient samples treated ex vivo. Here, we tested if CD46–ADC may have the potential to target MM-initiating cells (MM-ICs). CD46 expression was measured on primary MM cells with a stem-like phenotype. A patient-derived xenograft (PDX) model was implemented utilizing implanted fetal bone fragments to provide a humanized microenvironment. Engraftment was monitored via serum human light chain ELISA, and at sacrifice via bone marrow and bone fragment flow cytometry. We then tested MM regeneration in PDX by treating mice with CD46–ADC or the nonbinding control–ADC. MM progenitor cells from patients that exhibit high aldehyde dehydrogenase activity also have a high expression of CD46. In PDX, newly diagnosed MM patient samples engrafted significantly more compared to relapsed/refractory samples. In mice transplanted with newly diagnosed samples, CD46–ADC treatment showed significantly decreased engraftment compared to control–ADC treatment. Our data further support the targeting of CD46 in MM. To our knowledge, this is the first study to show preclinical drug efficacy in a PDX model of MM. This is an important area for future study, as patient samples but not cell lines accurately represent intratumoral heterogeneity.

## 1. Introduction

Multiple myeloma (MM) is an incurable plasma cell malignancy that is diagnosed in more than 35,000 Americans per year, and causes more than 12,000 deaths/year [1]. Options for treatment in the past two decades have substantially improved to include immunomodulatory drugs (IMiDs), proteasome inhibitors (PIs) and monoclonal antibodies (mAbs), mostly given in two, three or four drug combination regimens [2,3]. Unfortunately, none of these agents have proven to be curative, allowing the development of resistance after the sequential lines of therapy and ultimately leading to patient mortality. Importantly, preclinical studies in MM widely use cell line xenografts, but do not accurately represent intratumoral heterogeneity that is present in the human disease and therefore do not provide accurate means to test for curative potential of new drugs.

One way to improve therapeutic outcome is to identify and target novel antigens present on MM cell surface that are associated with or responsible for relapse. The first effective immunotherapy target for MM was CD38, initially exploited via daratumumab, a mAb initially used for relapsed/refractory disease and more recently incorporated into frontline. CD38 is an excellent target that is highly expressed on the surface of the bulk of malignant cells. However, not all MM cells are eliminated, and resistance develops with CD38 downregulation after prolonged treatment [4,5]. The effectiveness of later re-treatment with CD38-targeting antibodies after the development of resistance has been limited [6,7,8]. Recently, the FDA approved of two chimeric antigen T (CAR-T) cell therapies and one bispecific antibody (bsAb) that targets B cell maturation antigen (BCMA) [9,10,11]. BCMA is expressed exclusively on normal plasma cells and MM cells. Despite the excitement around anti-BCMA CAR-T and bsAbs, neither appear to be curative in MM, similar to the case when targeting CD38, and resistance eventually occurs. 

Another immunotherapy modality which has not yet reached its full benefit for patients with MM is antibody–drug conjugates (ADCs). ADCs combine antibody specificity with the delivery of highly cytotoxic chemotherapy, increasing selectivity by making the antibody a delivery vehicle that first recognizes the target and then binds and delivers the highly cytotoxic drug to malignant cells, while sparing target-negative tissues. MM is appealing for ADC therapy due to the entry of infused large-molecule drugs into malignant cells in the bone marrow (BM) through the sinusoidal endothelium. The most advanced ADC in MM was belantamab mafadotin, which also targets BCMA and received accelerated FDA approval. Unfortunately, this was later withdrawn due to its modest clinical benefits and ocular toxicities [12,13]. Still, the use of ADCs targeting alternative antigens besides BCMA may be a useful way to avoid cross-resistance in patients relapsing after CAR-T and bispecific antibodies. 

Currently, it remains our biggest challenge that MMs are not eliminated using our current treatments. This is true even when patients have undetectable levels of minimal residual disease (MRD) after treatment. This implies that MRD contains MM-initiating cells (MM-ICs) that are inherently resistant to current therapies. Towards the objective of seeking alternative antigens that are present on MM MRD cells, we reported the preclinical results for a novel ADC targeting the cell surface complement inhibitor CD46, showing encouraging potential in MM cell line xenografts [14]. An intriguing feature of this target is that the CD46 gene is located on chromosome 1q, which is frequently amplified in MM, detectable in one out of three patients at diagnosis and conveys a poor prognosis. We found that the cell surface expression level of CD46 was highest and CD46–ADC was most potent in patient MM cells with 1q+ [14]. Here, we sought to further study the potential of this novel agent by testing the hypothesis that CD46–ADC has unique potential to selectively eliminate 1q+ MM-ICs using patient-derived xenografts (PDXs). 

## 2. Material and Methods

### 2.1. Patient Samples

BM aspirates were collected from patients at the University of Colorado after informed consent was obtained and protocol was approved by the Institutional Review Board. The identification of patient information was replaced with allocating sequential numbers. Mononuclear cells (MNCs) were isolated from the samples via Ficoll density gradient centrifugation using SepMate Ficoll-Plaque tubes (StemCell Technologies, Vancouver, BC, Canada). Samples were cryopreserved in a freezing medium consisting of Iscove’s modified Dulbecco’s medium (IMDM), 45% fetal bovine serum (FBS), and 10% dimethylsulfoxide (DMSO) in an amount of 10 million cells/mL.

### 2.2. Antibody–Drug Conjugate Synthesis

A monomethyl auristatin F (MMAF) anti-tubulin payload was conjugated to anti-CD46 (CD46–ADC) as we described previously [14]. Briefly, human IgG1 23AG2 (aka YS5) was reduced using tris(2-carboxyethyl)phosphine (TCEP) at 37 °C for 2 h, purified using a Zeba spin column (Pierce/Fisher, Waltham, MA, USA), buffer-exchanged into PBS with 5 mM EDTA and incubated with mc-vc-pab-MMAF at room temperature for 1 h. Conjugation products were purified using the Zeba spin column (2×) to remove free MMAF and analyzed via HPLC using hydrophobic interaction chromatography with Infinity 1220 LC System (Agilent, Santa Clara, CA, USA). The drug to antibody ratio (DAR) was estimated via area integration using the OpenLab CDS v2.7 software (Agilent) (for CD46-ADC, DAR = 3.3). For the control, nonbinding human IgG1 (YSC10) was conjugated to the same linker–payload using the same method (DAR = 4.1).

### 2.3. Flow Cytometry

Flow cytometry was performed on FACSCelesta (BD) equipped with a high-throughput sampler. Results were analyzed using FlowJo v9 software (BD). To identify the profile of the MM population, extracellular surface marker stainings were performed using the following: anti-CD38-BB515 (HIT2), anti-CD138-BV421 (MI15), anti-CD45-BV510 (HI30), anti-CD19-BV786 (SJ25C1), anti-CD56-APC-R700 (NCAM16.2), anti-CD319-AF647 (235614) and lastly anti-CD46-AF647 (E4.3, labeling kit from Invitrogen, Waltham, MA, USA). All antibodies were purchased from BD Biosciences. Stained samples were washed and re-suspended in LIVE/DEAD Fixable Near-IR Stain (Thermo Fisher Scientific, Waltham, MA, USA) to determine cell viability, as described previously [15]. To identify primary myeloma cells that expressed high levels of Aldehyde dehydrogenases, we stained cells using the flow cytometer ALDEFLUOR™ Kit (Stemcell Technologies).

### 2.4. Patient Derived Xenografts

Research was approved by the Institutional Animal Care and Use Committee at the University of Colorado. Briefly, 2.5-month old NSG (NOD SCID gamma) mice were selected to be subcutaneously implanted with human fetal bone grafts (with femurs and tibias of 19–23-gestational week fetuses) as described, aiming to provide a human BM microenvironment to allow MM survival [16,17]. Six weeks post-surgery, primary myeloma samples (bulk MNCs from BM) were injected into the human BM grafts. Each primary cell sample was split evenly to inject 1 × 10^6^ cells into 3 or 4 mice. Engraftment was monitored weekly via the submandibular collection of serum for human light chains. CD46–ADC was administered via IV injection, with nonbinding IgG1 ADC used as a control, as free MMAF carries more toxicity when unconjugated to an antibody [18]. Mice were sacrificed at 30 weeks; mouse and human bone fragments were flushed for flow cytometry analysis. 

### 2.5. Human Light Chain ELISA

Immuno-nonsterile 96-well plates (Thermo Fisher) were coated with 1:400 dilutions of anti-human Kappa or anti-human Lambda free light chain antibodies (Abcam, Cambridge, UK) for 24 h. Plates were washed with PBS/Tween20 buffer and then blocked with BSA (Thermo Fisher) for 2 h at 4 °C. Mouse sera were diluted in a ratio of 1:20, plated and left overnight at 4 °C. After washing, secondary antibody mouse anti-human kappa/lambda AP (SouthernBiotech, Birmingham, AL, USA) was applied and incubated for 2 h at 37 °C. Plates were washed and a developing buffer containing diethanolamine (Thermo Fisher) and phosphatase substrate (SigmaAldrich, St. Louis, MI, USA) was added and incubated at 37 °C. Plates were read at 405 nm and analyzed using Versamax ROM v2.0.16 software.

### 2.6. Statistics

Statistics and figures were generated using Prism (GraphPad v10 Software). All data are presented as the mean and standard deviation. A two-tailed Student’s *t* test was used for comparing two means. When comparing more than two means, ANOVA was used with Tukey’s correction. Levels of statistical significance were labeled as follows: * *p* < 0.05, ** *p* < 0.01, *** *p* < 0.001 and **** *p* < 0.0001.

## 3. Results

### 3.1. CD46 Expression on Phenotypically Immature MM Cells

We evaluated whether or not CD46 is highly expressed on myeloma cells that are phenotypically “stem-like”. To assess the total clonal myeloma population via flow cytometry, we first characterized the expression of CD138, CD38 and other markers that differentiate myeloma cells (not shown: CD19, CD27 and CD45). Like normal adult stem cells, tumor-initiating cells are quiescent and toxin resistant. Aldehyde dehydrogenase (ALDH) 1A1 activity can be used to measure toxin resistance via flow cytometry (Figure 1A). Interestingly, we observed that the ALDH-high subpopulation in MM patients was increased in size in relapsed samples compared to samples from diagnosis (Figure 1B). The clonal MM cell subpopulations from five patients that exhibit high ALDH activity persistently express high levels of CD46 (Figure 1C). Next, we evaluated CD46 on the cell population that had been previously shown to possess MM-IC activity in vivo [19,20]. These cells exist in MM patient peripheral blood and express CD19 and CD22, are ALDH-high and also have a high expression of CD46 (Figure 1D). Thus, it appears that CD46 is persistently expressed in MM cells that are stem-like.

### 3.2. Treatment-Naïve Myeloma Cells Engraft Better in PDX 

To conduct an in vivo study of CD46–ADC on stem-like subpopulations of myeloma cells, we first piloted a PDX model of MM with human fetal bone grafts [17]. The efficiency of engraftment of myeloma samples in this model has been reported to be ~33% [21]. We hypothesized that relapsed disease would be more aggressive and therefore engraft better than newly diagnosed samples would. Thus, we performed an experiment testing three newly diagnosed samples versus three late-stage patient samples (three to four mice/per sample) (Table 1). We subcutaneously implanted bone grafts in the hindquarters of NSG mice. After surgery and 6 weeks of recovery, 1 × 10^6^ MNCs were injected directly into the grafts and mice were monitored using weekly serum human light chain ELISA (Figure 2A). We observed engraftment in 8/19 mice (42%) via restricted light chains matching the patients (Figure 2B). The study was terminated 6 weeks after cell injection, and flow cytometry of the bone grafts was used to confirm myeloma cell engraftment in mice with detectable light chain via ELISA at the time of sacrifice (Figure 2C). Surprisingly, the newly diagnosed samples were engrafted in 70% of mice (7/10) compared to 10% (1/10) mice from the relapsed/refractory samples (Figure 2D). There may be a fitness advantage present in treatment-naïve primary MM cells associated with patient treatment history and how it relates to the bone marrow microenvironment of fetal tissue; thus, newly diagnosed samples were used to study CD46–ADC. 

### 3.3. CD46–ADC Prevents the Engraftment of MM PDX

We next tested CD46–ADC for the ability to prevent primary myeloma cell regeneration and expansion. Each patient sample was injected into 3–4 mice, and split between two treatment groups of 13 mice each (Table 1). During the 2 weeks after primary cell injection, one group was treated with CD46–ADC and one was treated with the nonbinding control–ADC in four doses at 5 mg/kg IV, twice per week (Figure 3A). Again, human light chain ELISA was performed weekly to determine MM engraftment and monitor progression, until the sacrifice of the mice 6 weeks after the first treatment injection. A significant decrease in engraftment was seen with CD46–ADC treatment as early as week three (*p* = 0.038) and persisted through to week eight, at endpoint analysis (*p* = 0.005) (Figure 3B,C). Likewise, when looking at the BM involvement of those that engrafted via flow cytometry, we noticed that BM from mice treated with CD46–ADC had close-to-zero BM involvement (Figure 3D,E). Among all mice, we observed only 23% engraftment (3/13) via ELISA in those treated with CD46–ADC compared to 69% engraftment (9/13) for those treated with the nonbinding control (Figure 3F). Overall, engraftment was almost completely abrogated by CD46–ADC, suggesting that targeting CD46 can prevent disease regeneration. 

## 4. Discussion

The clinical reality that MM is an incurable disease infers that MM-initiating cells (MM-ICs) present during remission are inherently resistant to current therapies. However, cell line xenograft models are relied upon in MM preclinical research even though they are more homogeneous than primary tumors. There is controversy in the field regarding whether or not MM-ICs express unique cell surface markers or are resistant genetic subclones that cannot be distinguished from the major diseased clone [22]. Regardless of the answer to that question, persistent MM-ICs ultimately underlie persistence and ultimate relapse. Thus, it follows that that MM-ICs must be successfully targeted by new therapies to achieve a cure. In theory, an immunotherapy could be curative in MM if the target antigen is expressed in MRD/MM-ICs. However, the current immunotherapy targets in MM consist of the differentiation antigens CD38 and BCMA that may not be expressed on MM-ICs, and therapies targeting them have not been curative. Thus, we sought to extend our prior preclinical work with CD46–ADC by testing human primary MM samples in patient-derived xenografts. 

CD46 is an intriguing candidate with which to target MM-ICs, because (1) it is a complement inhibitor that may help MM-ICs survive and (2) a subset of high-risk MM patients have a genomic gain of CD46 on chromosome 1q (1q+). As the CD46 gene is located on chromosome 1q, a clinical 1q FISH (fluorescence in situ hybridization) probe that is on a widely used FISH panel utilized for risk stratification may provide a useful surrogate marker for CD46 overexpression [14,23]. As found via FISH, 1q+ occurs in approximate one of three newly diagnosed patients with MM and it becomes more common in relapsed disease (~80% of patients) [24]. Previously, we found that the cell surface expression level of CD46 was highest and CD46–ADC was most potent in patient MM cells with 1q+ [14]. If CD46 is highly expressed on 1q+ MM-ICs, and CD46–ADC can be demonstrated to eliminate disease with a patient-relevant approach, there would be sound rationale to design a clinical trial in the remission setting in attempt to achieve cure.

Here, we sought to determine if CD46 is expressed on MM-ICs and furthermore an appropriate target for therapy development by utilizing a patient sample- (not cell line-) based approach. We found that CD46 is highly expressed on MM cells that are phenotypically stem-like, although we did not test the activity of CD46–ADC against these cells directly in vitro. Since in vivo studies with patient-derived xenografts are the gold standard for determining tumor-initiating cell potential, we studied the effect of CD46–ADC in those models. Although we found that the obtainable PDX model did not support prolonged engraftment, CD46–ADC did abrogate primary MM cell engraftment, suggesting that this agent can target MM-ICs. Without long-term engraftment, we were unable to characterize the small number of MM cells that evaded CD46–ADC treatment, including effects of CD46–ADC on ALDH expression. Other alternative targets for immunotherapy in addition to BCMA and CD38 include GPRC5D, FCRH5 and ICAM1 [25,26,27]. Future studies of novel immunotherapies in PDX may help determine which target antigens have potential to eliminate MM-ICs. 

Research on MM-ICs has been limited by the challenges of perpetuating myeloma primary samples in PDX. These models utilizing nude mice with or without fetal bone chips have been used to study MM-ICs, but to our knowledge our study is the first to use them to test a potential MM therapeutic. Humanized mouse models with knock-in genes for key hematopoietic growth factors have improved the potential for using PDX in preclinical drug development for MM. NSG-SGM3s produce SCF, GM-CSF and IL-3 and have been good for myeloid engraftment, but do not help with lymphoid engraftment [28]. However, an important advance in developing a humanized mouse model for myeloma engraftment occurred with the report of the MISTRG6 model, a mouse with human IL-6 that showed successful engraftment in >80% primary MM samples [29]. However, this model has not been generally available and thus we chose the bone graft model. Despite its challenges, we did find surprisingly high engraftment efficiency in newly diagnosed, treatment-naïve myeloma samples that were used, albeit with limited longevity. Serial transplants do not appear feasible in the fetal bone chip model to truly demonstrate MM-IC capacity. These limitations may be overcome in future studies utilizing newer-genetic knock-in PDX models of MM such as MISTRG6.

Currently, most of the recent drug developments for relapsed/refractory MM have focused on T cell-re-directing therapies, and the ADC target with the most clinical experience is BCMA. Thus, target antigen-dependent and T cell-mediated mechanisms of resistance may be overlapping and become predominant in late-stage disease. CD46 is an emerging new target. A phase I clinical trial for CD46–ADC (FOR46, Fortis Therapeutics) is now complete in patients with relapsed or refractory MM (NCT03650491), showing single-agent activity with partial response or better observed in approximately one-third of patients [30]. This agent is also being developed for the treatment of prostate cancer [31]. Our data further support the targeting of CD46 in MM by supporting its potential to abrogate disease regeneration.

## 5. Conclusions

Developing therapies with the ability to target MM-ICs is vital to advancing myeloma patient care through the prevention of disease relapse. To the best of our knowledge, this study is novel in that it is the first to demonstrate preclinical drug efficacy in a MM PDX model and confirm CD46 expression on the surface of myeloma stem-like cells. By establishing CD46 as an effective ADC target for preventing primary MM cell engraftment, we further support the notion that CD46 is an emerging target in multiple myeloma worthy of continued study.

## Figures and Tables

**Figure 1 cancers-15-05335-f001:**
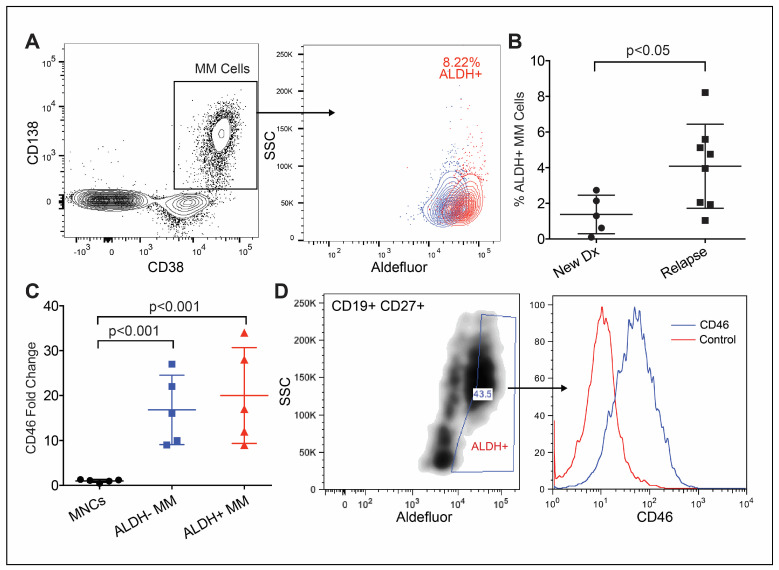
CD46 is highly expressed on primary multiple myeloma cells with immature characteristics. (**A**) Representative primary MM bone marrow sample gated on kappa-restricted (not shown) CD38+/CD138+/clonal MM cells, followed by ALDH+ (red) and subtracted by the DEAB control (blue). (**B**) A higher percentage of ALDH-high cells in samples from relapsed patients (*n* = 8) compared to that in newly diagnosed patients (*n* = 5). (**C**) ALDH+ MM cells with significantly higher CD46 compared to that of normal MNCs (*n* = 5). (**D**) Peripheral blood sample from patient with clonal CD19+/CD37+/ALDH+ MM progenitors with a high expression of CD46. Error bars—SD.

**Figure 2 cancers-15-05335-f002:**
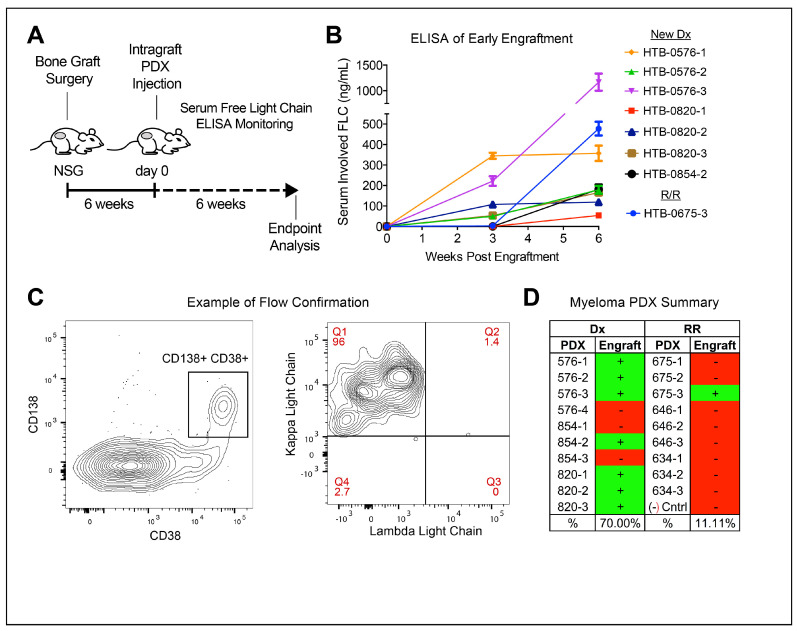
Samples from newly diagnosed MM patients engraft better than do those from patients with relapsed/refractory MM. (**A**) NSG mice harboring human fetal bone grafts injected with patient samples, and clonal human free light chain detected in serum of engraftment mice via ELISA for 6 weeks followed by endpoint analysis. (**B**) Serum ELISA resulting in mice showing detectable levels of human light chain that matched the kappa or lambda restriction of the patient sample of origin. (**C**) Detection of CD38+, CD138+ (left panel) and light chain-restricted (right panel) human myeloma cells via flow cytometry of cells isolated from bone graphs with a detectable serum-free light chain. (**D**) Myeloma engraftment in 8/19 (42%) and samples selected from newly diagnosed (Dx) patients being engrafted more frequently than samples from relapsed/refractory (RR) patients. Error bars—standard deviation.

**Figure 3 cancers-15-05335-f003:**
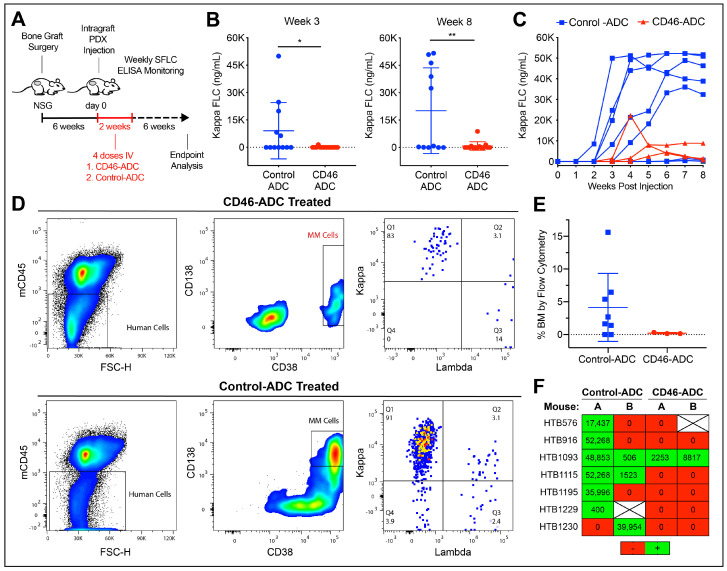
In vivo treatment with CD46–ADC decreasing multiple myeloma engraftment in patient-derived xenografts. (**A**) Fetal bone grafts subcutaneously implanted in NSG mice, injected with patient samples at day 0, then treated with CD46– or nonbinding control–ADC. (**B**,**C**) Weekly human serum-free light chain (SFLC) ELISA from mice (undetectable levels omitted in (**C**)). (**D**,**E**) Bone marrow involvement of human MM cells at week 8 measured via flow cytometry, as representative examples (**D**) and as a group (**E**), showing very-low-level engraftment in mice treated with CD46–ADC. (**F**) Overall engraftment according to SFLC ELISA results for each sample (green = engrafted red = no engraftment). HTB—hematology tissue bank, * *p* < 0.05, ** *p* < 0.01.

**Table 1 cancers-15-05335-t001:** Clinical characteristics of multiple myeloma patients selected for use in patient-derived xenograft studies.

	DISEASE STATE	BM MM%	IG	PRIOR TREATMENT	FISH
**HTB-0576**	New diagnosis	40–50%	KLC	None	Hyperdiploid, 13q-, uncharacterized IgH translocation, 11q+
**HTB-0820**	New diagnosis	95%	KLC	None	t(8;22), 1q+, 13q-, 11q+
**HTB-0854**	New diagnosis	80%	KLC	None	t(4;14), 1p-, 12-
**HTB-0675**	Relapse	90%	LLC	RVD, ASCT, Rm, VDT-PACE, ASCT, Pom, CarPomD, Panda/RD	1q+, t(11;14)
**HTB-0634**	Relapse	50–60%	LLC	VD, ASCT, RD, VD, KD, Ixa/Dex	11q+
**HTB-0646**	Relapse	80%	LLC	VD, VDPACE, ASCT, KRD, Benda-RD, Elo-RD, Dara	1q+ 13q- 1p-, t(4;14)
**HTB-0916**	New diagnosis	80%	KLC	None	t(11;14)
**HTB-1093**	New diagnosis	90%	KLC	None	t(8;14), 1q+, 13-
**HTB-1115**	New diagnosis	70%	KLC	None	t(11;14), 1q+, 11q+
**HTB-1195**	New diagnosis	90%	KLC	None	t(8;22), t(14;16), 1q+, +11, +13, +17
**HTB-1229**	New diagnosis	40%	KLC	None	+11q, +17
**HTB-1230**	New diagnosis	75%	KLC	None	Hyperdiploid, 1q+, 13q-

Abbreviations: ASCT: autologous stem cell transplant. Benda-RD: bendamustine, lenalidomide and dexamethasone. BM MM%: bone marrow involvement. CarPOM-D: carfilzomib, pomalidomide and dexamethasone. Dara: daratumumab. Elo-RD: elotuzumab, lenalidomide and dexamethasone. F: female. Ig: Immunoglobulins. Ixa/Dex: Ixazomib and dexamethasone. KD: carfilzomib/dexamethasone. KLC: kappa light chain. KRD: carfilzomib, lenalidomide and dexamethasone. LLC: lambda light chain. M: male. Panda/RD: lenalidomide and low-dose dexamethasone. Pom: pomalidomide. RD: lenalidomide and dexamethasone. Rm: Revlimid maintenance. RVD: lenalidomide, bortezomib and dexamethasone. VD: bortezomib and dexamethasone. VDT-PACE: bortezomib, dexamethasone, thalidomide, cisplatin, doxorubicin, cyclophosphamide and etoposide.

## Data Availability

The data presented in this study are available on request from the corresponding author.

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
