# Peer review of "CD46–ADC Reduces the Engraftment of Multiple Myeloma Patient-Derived Xenografts"

_cancers, 2023, doi:10.3390/cancers15225335_

Round 1
Reviewer 1 Report
Comments and Suggestions for Authors
The study "CD46-ADC Reduces the Engraftment of Multiple Myeloma Patient-Derived Xenografts" was well executed and conducted. The results of the study were consistent and statistically significant. The authors provided a comprehensive interpretation of the findings and suggested potential implications of the results for future research. The study is a significant contribution to the field of multiple myeloma research. It is encouraged for researchers to include limitations and future plans for this study as minor comments.
Author Response
Response: Thank you for your helpful evaluation of our manuscript. In response to your comment, we have expanded the DISCUSSION section to include more on limitations and future plans, including the evaluation of other emerging myeloma targets such as BCMA, GPRC5D, and FcRH5 and their ability to eliminate MM initiating cells. We agree that our study has some limitations including the PDX model not supporting prolonged engraftment, hampering our ability to characterize the few cells that evaded in vivo CD46-ADC treatment (page 9, paragraph 2).
Reviewer 2 Report
Comments and Suggestions for Authors
It is a very interesting and meaningful study. The authors for the first time verify the CD46 is a promising target for preventing primary MM cell engraftment by CD46-ADC in a MM PDX model. The manuscript was well organized, and the results were well presented. It was a paradigm of therapeutic research of a new target. It should be accepted with some minor revision.
1. Could the authors give more discussion on why the newly diagnosed samples engrafted in 70% of mice (7/10) compared to 10% (1/10) mice from the relapsed/refractory samples (Figure 2D)?
2. How about the treatment efficiency of free drug (Monomethyl auristatin F)? Is free Monomethyl auristatin F necessary as a control group?
3. What is the influence of the CD46-ADC on the ALDH activity of MM?
Comments on the Quality of English LanguageMinor editing of English language required
Author Response
- Could the authors give more discussion on why the newly diagnosed samples engrafted in 70% of mice (7/10) compared to 10% (1/10) mice from the relapsed/refractory samples (Figure 2D)?
Response: Thank you for posing this important question. We agree that it is interesting that newly diagnosed samples engrafted more often compared to those containing relapsed/refractory disease. The patient cohort for this study contained a range of cytogenetics, risk levels, bone-marrow involvement, and ages rendering these factors unlikely related to engraftment potential. However, there are several other possibilities for influencing the result. Each relapse/refractory patient in this study had received chemotherapy, an autologous stem-cell transplant, and multiple anti-myeloma agents. It is plausible the naïve bone marrow microenvironment present in fetal bone chips is less similar to that of a relapsed refractory patient. In addition to micro-environment alterations, the clonal evolution of MM over the course of treatment likely influences the ability to engraft within fetal bone chips. We have updated the RESULTS section of the manuscript to reflect these points (page 5, paragraph 1).
- How about the treatment efficiency of free drug (Monomethyl auristatin F)? Is free Monomethyl auristatin F necessary as a control group?
Response: Thank you for bringing up this point. MMAF is a newer generation auristatin that was designed to be more active when delivered inside the cell via antibody as compared to the untargeted form. Free MMAF has high off-target toxicity in vivo that are not relevant to the targeted delivery modality. Given these drug characteristics combined with the lower engraftment potential of the PDX model, we decided a non-binding ADC was the best control. We have updated the methods section to reflect this point and added an appropriate citation for the toxicity profile of free MMAF (Doronina et al. 2006 Bioconjug Chem) (page 3, paragraph 4).
- What is the influence of the CD46-ADC on the ALDH activity of MM?
Response: Thank you for raising this important question. We did not directly measure the influence of CD46-ADC on the ALDH activity of primary myeloma ex vivo for this manuscript. We have noted this limitation in the revised manuscript in the DISCUSSION section (page 9, paragraph 2).
Reviewer 3 Report
Comments and Suggestions for Authors
The novelty is incremental at best, compared to previous findings from the same group of Authors:
Antibody-drug conjugate targeting CD46 eliminates multiple myeloma cells. Sherbenou DW, Aftab BT, Su Y, Behrens CR, Wiita A, Logan AC, Acosta-Alvear D, Hann BC, Walter P, Shuman MA, Wu X, Atkinson JP, Wolf JL, Martin TG, Liu B. J Clin Invest;126:4640-4653. doi: 10.1172/JCI85856.
Comments on the Quality of English Language-
Author Response
Response: Thank you for the comment. To our knowledge, our manuscript is the first to demonstrate preclinical drug efficacy in a myeloma patient-derived xenograft model. As patient tumors are more heterogeneous than cell lines, we believe this is an important consideration for the future preclinical development of novel agents for multiple myeloma. Furthermore, our study is the first to evaluate and confirm CD46 expression on the surface of myeloma stem-like cells. Our results advance the notion that CD46 is a promising target in multiple myeloma by functionally demonstrating its ability to abrogate primary MM cell engraftment. To address your comment on this, we have made revisions to the SIMPLE SUMMARY, ABSTRACT, INTRODUCTION (page 2, paragraphs 1 and 4) and DISCUSSION (page 8, paragraph 1, and page 9, paragraph 2 and 3) and CONCLUSION (page10, paragraph 1) sections.
Round 2
Reviewer 3 Report
Comments and Suggestions for Authors
-
Comments on the Quality of English Language-